# Distribution and evolution of the LysR-type transcriptional regulators of the *Salmonella* genus

Dávila S.[1], Rivera-Ramírez A.[2], Gama-Martínez Y.[2], Orozco R.[2], Hernández V. M.[2], Hernández-Lucas I.[2]*

**1** Centro de Investigación en Dinámica Celular, Universidad Autónoma del Estado de Morelos, Cuernavaca, Morelos, México, **2** Departamento de Microbiologia Molecular, Instituto de Biotecnología, Universidad Nacional Autonoma de México, Cuernavaca, Morelos, Mexico

☯ These authors contributed equally to this work.

\* ismael.hernandez@ibt.unam.mx

## Abstract

LysR-type transcriptional regulators (LTTRs) are one of the most abundant transcriptional regulators in nature and are involved in multiple essential biological process in bacteria. In this work we show that LTTRs are highly abundant in *Salmonella enterica* subspecies *enterica* serovars able to infect humans (*S.* Typhi), whereas the number of LTTRs decreases substantially in reptile commensals (*Salmonella enterica* subspecies *arizonae, diarizonae, houtenae, indica, salamae* and *Salmonella bongori*). In addition, it is also reported the presence of a *Salmonella* LTTR core, LTTRs exclusive to the *Salmonella* genus and LTTRs like CysB that is widely distributed in the *Enterobacteriales* order and their orthologous sequences separate clades at the genus level, suggesting that CysB has evolved in parallel with the corresponding lineage. Therefore, there are LTTRs that evolved as part of the core of microorganisms that provide essential genetic functions to the cell, as well as a LTTR accessory pool that provides different capabilities to specific microorganisms for survival in nature and in different environments of the host.

## Introduction

The *Salmonella* genus consists of two species, *Salmonella bongori* and *Salmonella enterica*. *S. enterica* is divided into six subspecies: *arizonae, diarizonae, enterica, houtenae, indica* and *salamae.* In the case of the *S. enterica* subspecie *enterica* (*S. enterica*), over 2600 different serotypes have been described and most of them are related to human and animal infections. *S. enterica* causes infections mainly in warm-blooded animals [1–3], while subspecies *arizonae, diarizonae, houtenae, indica* and *salamae* are prevalent in reptiles, and sporadically cause infections in humans. The second species of *Salmonella, S. bongori,* is more common in cold-blooded animals, especially reptiles, and in the environment, but can also infect humans at very low

**Data availability statement:** All relevant data are within the manuscript and its Supporting Information files.

**Funding:** This work was supported by grants from: Dirección General de Asuntos del Personal Académico (DGAPA/UNAM) (IN203621-IN202224) and Consejo Nacional de Humanidades, Ciencias y Tecnologías (CONAHCYT) (CF-2023-I-2079) to I.H.L. The funders had no role in study design, data collection and analysis, decision to publish, or preparation of the manuscript.

**Competing interests:** The authors have declared that no competing interests exist.

rates [4]. The limited pathogenicity and poor ability of *S. bongori*, *S. arizonae, diarizonae, huotenae, indica* and *salamae* to invade host cells is due to modifications in some important virulence factors of *Salmonella* pathogenic island I (*invB, invF and invI*) and *Salmonella* pathogenic island II (*ssaE, ssaG and ssaH*). The great majority of human infection by these subspecies are related to a previous depression of the immune system [4,5].

It is estimated that *Salmonella* causes from 200 million to over 1 billion infections annually worldwide, with 93 million cases of gastroenteritis, 155,000 deaths, and 85% of illnesses which are food-linked [6], constituting a serious global health problem.

To survive in nature and persist within humans or animals, *Salmonella* relies on its genetic arsenal that includes transcriptional regulators, like those that belong to the LTTR family. LTTRs are DNA-binding proteins involved in transcriptional regulation of genes for central metabolism, amino acid biosynthesis, cell division, fermentation, photosynthesis, nitrogen fixation, oxidative stress, quorum sensing, motility, transport of $Na^+$, symbiosis, and virulence. These transcriptional factors are widely distributed in nature, predominantly in Gram-negative organisms and are constituted by 300–350 residues including a DNA-binding domain (DBD) at the N-terminus which interacts with the promoter region. A long linker helix involved in oligomerization that connects the DBD with the C-terminal periplasmic binding domain, also called the regulatory domain (RD), where an effector binds to modulate transcriptional expression of the LTTR target genes [7].

LTTRs are present in eukaryotic cells, archaea, and bacteria. Furthermore, these proteins are one of the most abundant transcriptional factors in bacteria [7]. *S.* Typhi contains 41 LTTRs in its genomic organization (This work). Previously, we described the function of three LTTRs in *S.* Typhi: LeuO is involved in the regulation of detoxification, porin synthesis and in the positive control of the CRISPR-Cas system as well as in virulence [8–11]; STY0036 has a role in porin synthesis, bile resistance and in genetic transformation [12,13]; STY2660 participates in porin synthesis, bile resistance and motility [14]. These results show the relevance of LTTRs in *Salmonella*. In this work, we demonstrate that in general LTTRs are more abundant in *Salmonella* serovars able to infect humans (*S.* Typhi) as compared with *Salmonella* subspecies that are reptile commensals. Furthermore, there is a *Salmonella* LTTR core, LTTRs exclusively of the *Salmonella* genus, and LTTRs widely distributed in the *Enterobacteriales* order. Remarkably, there are LTTRs that evolve as the genome core of many microorganisms providing essential genetic functions to the cell as the highly conserved and ubiquitous LTTR CysB, that plays a key role in controlling expression of genes encoding both transporters of sulfur-containing compounds as well as enzymes involved in incorporating the sulfur into cysteine [15], in addition there is a LTTR accessory pool that provide different capabilities to specific microorganisms for survival in nature and in different environments of the host.

## Materials and methods

### LTTR identification in *Salmonella* and bacterial genomes

The PROSITE database was used to obtain LTTRs from the *S.* Typhi CT18 genome (ID 220341) (https://prosite.expasy.org/PS50931). The conserved domain accessions

PF0126 and PF03466 were applied to verify the *S*. Typhi LTTR authenticity. Furthermore ClustalW ([https://www.genome.jp/tools-bin/clustalw](https://www.genome.jp/tools-bin/clustalw)) sequence alignment showed that all the LTTRs identified in *S*. Typhi CT18 genome contain the winged helix DNA binding domain, the linker helix domain, and periplasmic binding domain characteristic of the LTTR family [7]. The resultant 41 LTTRs of *S*. Typhi (Fig 1) were used as input query sequences to find orthologous proteins in prokaryotic genomes contained in the UniProt database. LTTR sequences were selected only if they shared ≥90% amino acid identity with the LTTRs of *S*. Typhi. Using this filtering strategy *S*. Typhi LTTR orthologs were identified in 2052, 100, 523, 892, 449, 34 and in 491 genomes of *S. enterica*, *S. bongori*, *S. arizonae*, *S. diarizonae*, *S. houtenae*, *S. indica* and in *S. salamae* respectively. Thus, *S*. Typhi LTTRs orthologous sequences in 4541 genomes belonging to the *Salmonella* genus were identified. Orthologous LTTRs sequences of *S*. Typhi CT18 from other bacterial phyla, mainly belonging to the *Enterobacteriales* order were also obtained using this filtering strategy. The LTTR data obtained was used to generate

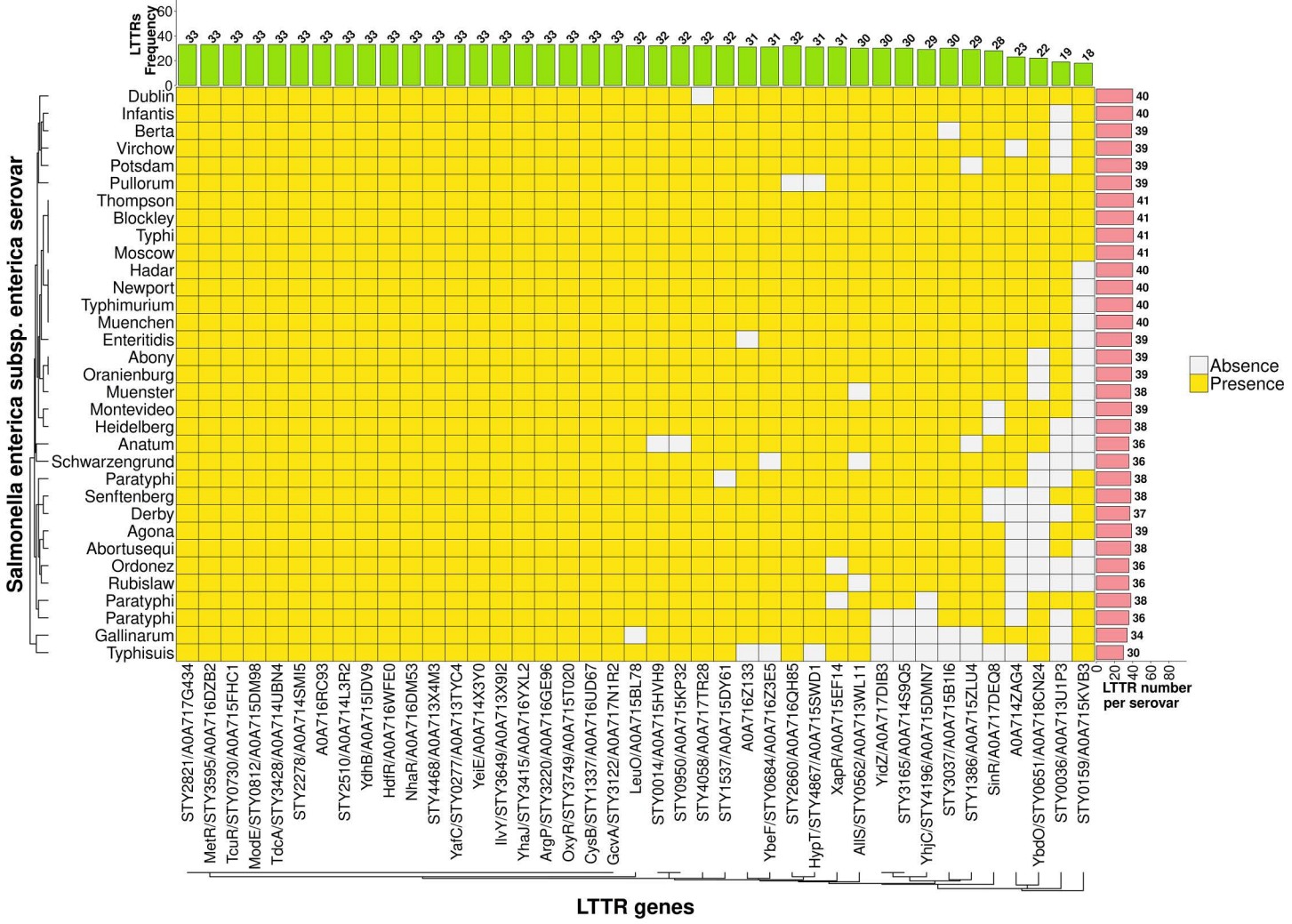

**Fig 1. LTTR distribution in *Salmonella enterica* subspecies *enterica*.** LTTRs were obtained from the *S.* Typhi CT18 genome, and its orthologs in *S. enterica* subspecies *enterica* with an identity percentage equal to or greater than 90% in the entire protein were selected from the UniProt database and are represented by yellow filled squares..

Figs 1-5. In Table 1 supplemental material in S1 Table we include the assembly accessions of the genomes utilized to generate Figs 1 to 5.

## LTTR distribution in *Salmonella* and in other *Enterobacteriales* genomes

The phylogenetic analysis of LTTRs in *Enterobacterales* genomes was conducted using the JolyTree pipeline. First, pair-wise genomic distances were estimated with Mash, calculating k-mer-based dissimilarity approximating the p-distance. This value was corrected to evolutionary distance using the F81 model to account for nucleotide composition. The distance matrix was then used to infer the phylogenetic tree under the Balanced Minimum Evolution (BME) criterion with FastME. Branch confidence was assessed with REQ based on quartet consistency with the distance matrix. Finally, the resulting tree was manipulated and visualized using the ggtree package in R version 4.5.1, to generate Fig 4.

## Phylogenetic relationships of bacteria based on the LTTR CysB

To determine the phylogeny of the LTTR CysB in the broader context of bacterial species, a phylogenetic tree of these organisms based on CysB proteins was generated. CysB protein sequences belonging to the identified orthologous groups were aligned using MAFFT v7.505 [16], with default parameters to generate high-quality multiple sequence alignments. The resulting alignments were then used to infer phylogenetic relationships with IQ-TREE v2.0.7 [17]. Phylogenetic CysB reconstruction was performed under the ModelFinder Plus option (*-m MFP*) to automatically select the best-fit substitution model [18]. Branch support was assessed using both approximate Bayesian computation (*-abayes*) and ultrafast bootstrap with 1,000 replicates (*-B 1000*). The CysB tree was edited and visualized with FigTree (v1.4.2), to generate Fig 5.

# Results

## Distribution of LTTRs in the *Salmonella* genus

LTTRs are widely distributed in Gram-negative organisms including the *Salmonella* genus. In this work we focus on the LTTRs of *S.* Typhi, their identification, distribution, and presence of their orthologs in *Salmonella* and bacterial genomes. The main reason for this is that these transcriptional factors are involved in motility, bile resistance, porin synthesis, macrophage, liver and spleen replication, intestinal epithelial cell invasion, biofilm formation and in other fundamental steps of pathogenesis [7]. Therefore, the study of the *S.* Typhi LTTRs can help to know which of these transcriptional factors are exclusive of this pathogen to address its function in relation to prevention and control of typhoid fever. According to WHO (World Health Organization) data, approximately 11–21 million typhoid fever illnesses occur annually worldwide, accounting for 0.12 to 0.16 million deaths [19]. Thus, *S.* Typhi constitutes a significant global health concern. Remarkable this human pathogen contains a high number of LTTRs within *S. enterica* subspecies *enterica* (Fig 1), suggesting a relevant role of LTTRs in *Salmonella* virulence.

The *S.* Typhi genome contains 41 LTTRs. The genome analysis of *Salmonella enterica* subspecies *arizonae, diarizonae, houtenae, indica* and *salamae* shows the presence of 26, 30, 31, 31, and 34 LTTRs respectively. *S. bongori* contained 29 LTTRs in its genome organization. (Fig 2).

The LTTR numbers suggest a clear trend towards the reduction of these transcriptional factors in *Salmonella* reptile commensals such as *S. enterica* subspecies *arizonae, diarizonae, houtenae, indica, salamae* and in *S. bongori,* (Fig 2). Thus, these bacteria are able to adapt into the host or in the environment with a reduced number of LTTRs.

These results show clearly that the LTTR numbers in the distinct *Salmonella* groups varies considerably, thus in specific groups such as *S.* Typhi, LTTRs are abundant, constant and are preserved, while in other *Salmonella* groups LTTRs decrease considerably.

The distribution analysis of LTTRs of the *Salmonella* genus also demonstrated that there is a *Salmonella* LTTR core of 22 LTTRs present in *S. enterica* (*S.* Typhi), *arizonae*, *diarizonae*, *houtenae*, *indica*, *salamae* and in *S. bongori*. This LTTR

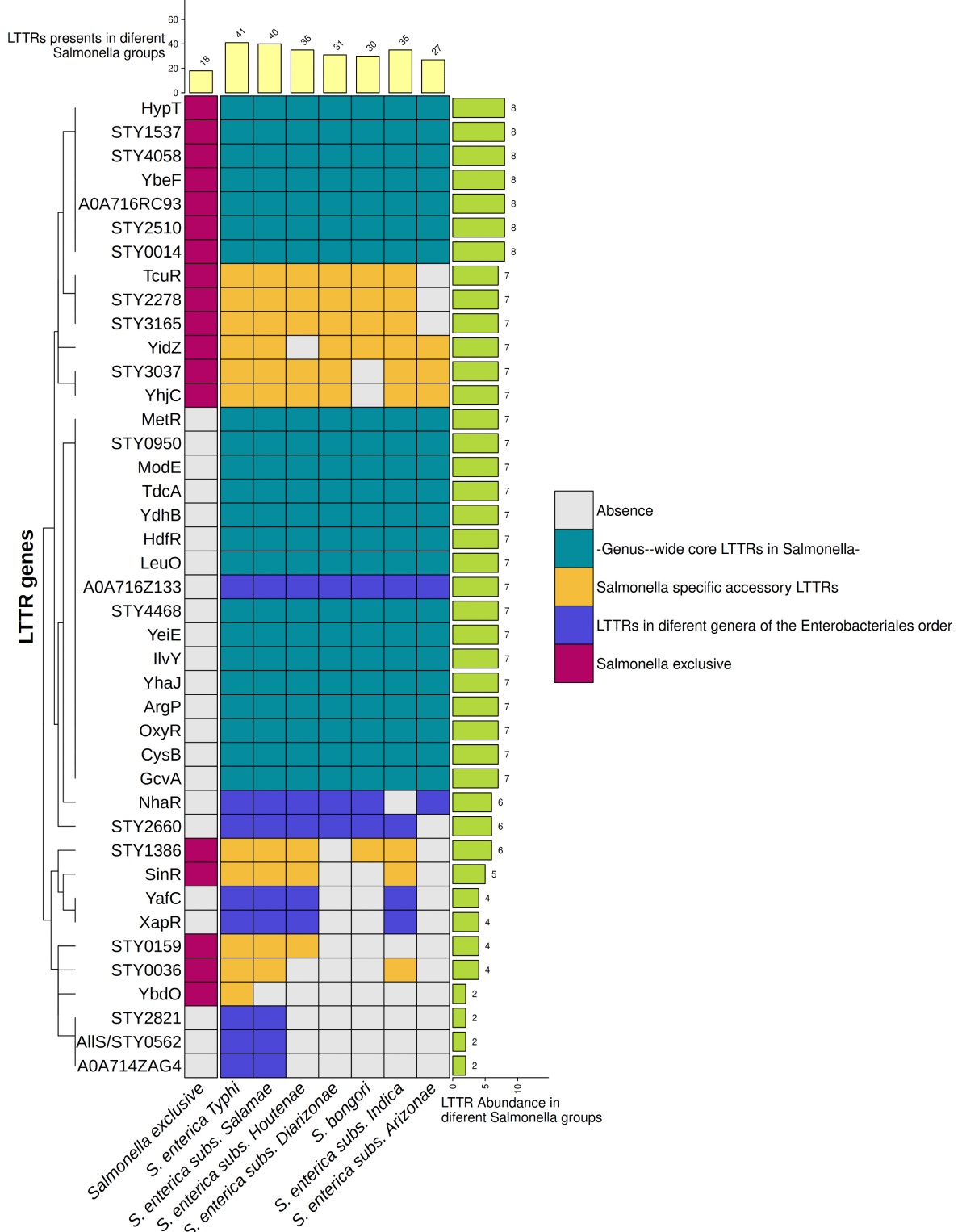

**Fig 2. LTTR distribution in the *Salmonella* genus.** LTTRs in *Salmonella* are shown in filled squares. The LTTR core is represented in green squares, the exclusives LTTRs of the *Salmonella* genus are in red squares, the *Salmonella* specific accessory LTTRs are in yellow squares and LTTRs present in different bacteria genera of the *Enterobacteriales* order are in blue squares. The more frequent LTTR number among the genomes analyzed is presented.

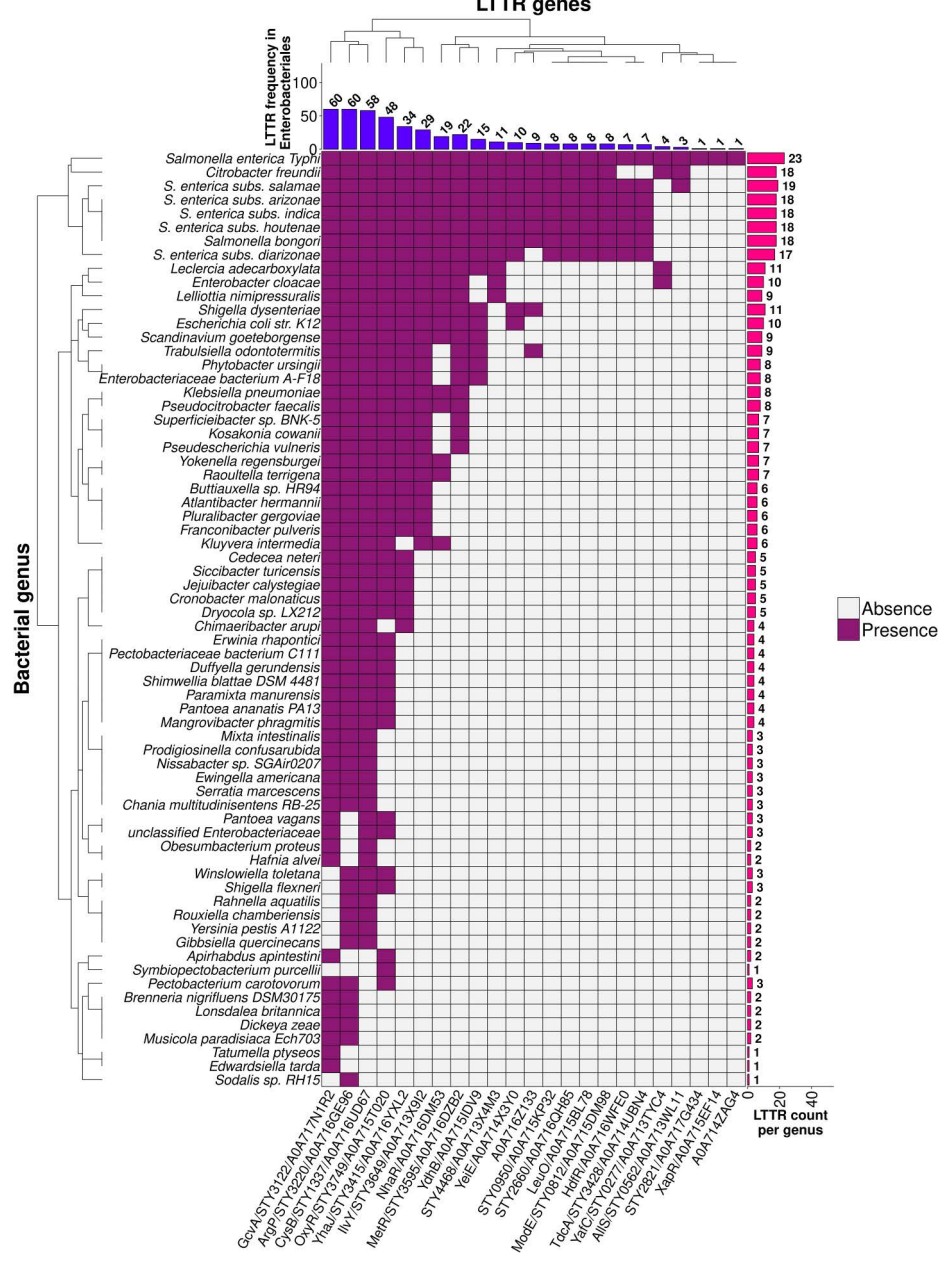

**Fig 3. LTTR distribution in *Enterobacteriales*.** LTTRs from the *S.* Typhi CT18 genome and its orthologs in *Enterobacteriales* with an identity equal to or greater than 90% over the entire protein were obtained from the UniProt database and are represented by purple filled squares. The table also shows the number of LTTRs in each representative taxonomic group as well as the number of genera analyzed.

core includes hypothetical proteins without assigned function (STY0950, STY4468, STY2510, STY0014, A0A716RC93, YbeF, STY4058, STY1537 and LTTRs involved in the control transport, synthesis, and catabolism of amino acids (MetR, CysB, GcvA, ArgP, IlvY) [20–24], porin synthesis (LeuO) [25], oxygen stress response (OxyR), aromatic compound degradation (YhaJ) [26], regulation of purine transport (YdhB) [27], sulfite tolerance (YeiE) [28], regulation of molybdenum metabolism (ModE) [29], flagella control (HdfR) [30], transport and metabolism of threonine-serine (TdcA) [31] and in the hypochlorous acid defense mechanism (HypT) [32].

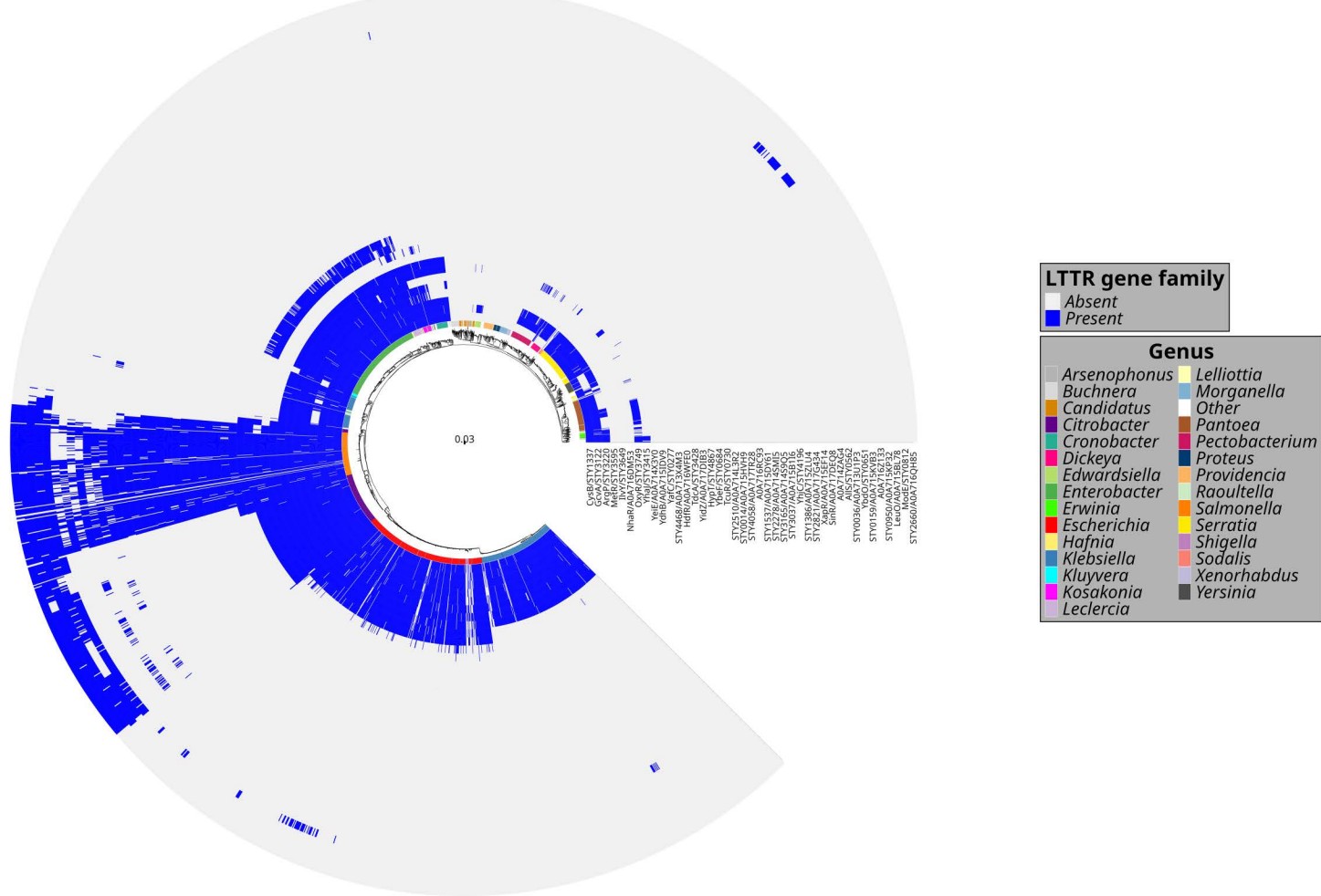

**Fig 4. LTTR in the Enterobacteriales order.** A phylogenetic tree showing LTTR distribution in different members of the *Salmonella* genus and in other *Enterobacteriales*..

(Fig 2). Therefore, the regulation of these essential biological process by these LTTRs is fundamental in the *Salmonella* genus.

Another finding of this investigation is that the distribution analysis of *Salmonella* LTTRs shows the presence of 18 LTTRs exclusive to the *Salmonella* genus (STY1386, STY2510, STY0014, A0A716RC93, STY2278, YbeF, STY4058, YidZ, STY1537, STY3037, STY3165, HypT, TcuR, SinR, STY0036, STY0159, YbdO and YhjC). Some of these LTTRs are part of the LTTR core. These regulators are not present in other bacterial genera considering an identity percentage equal to or greater than 90% with respect to the complete proteins of *S.* Typhi. 18 of these LTTRs were detected in *S.* Typhi, 16 in *S. salamae*, 14 in *S. houtenae*, 13 in *S. diarizonae*, 14 in *S. indica,* 10 in *S. arizonae*, and 12 in *S. bongori*. Interestingly most of these LTTRs are hypothetical proteins. The fact that these LTTRs are not present in other bacterial pathogens suggest that each pathogen acquires LTTRs for its specific way of life (Fig 2).

**LTTRs of the *Salmonella* genus are widely distributed in other *Enterobacteriales***

*S.* Typhi contains 23 LTTRs present in the *Enterobacteriales* order. These LTTRs correspond to the hypothetical proteins: STY0950, STY2660, STY4468, A0A716Z133, A0A714ZAGA4, STY2821 and MetR, CysB, GcvA, OxyR, ArgP, YhaJ, IlvY,

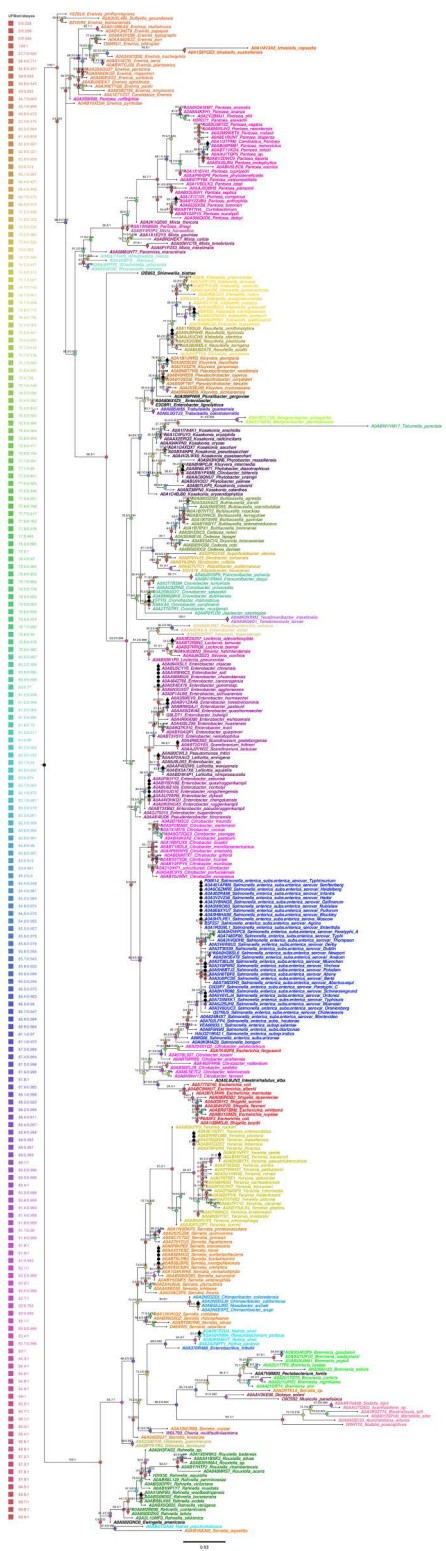

**Fig 5. The LTTR CysB evolve as the core genome of *Enterobacteriales*.**

YdhB, ModE and AllS/STY0562 that have been characterized in different bacteria [20–24,26,27,29,33]. The remaining LTTRs widely distributed in *Enterobacteriales* are involved in sulfite tolerance (YeiE) [28], resistance to nonvolatile compounds (YafC) [34], porin regulation (LeuO) [8], response to osmolarity (NhaR) [35], flagella control (HdfR) [30], xanthosine metabolism (XapR) [36], and transport and metabolism of threonine-serine (TdcA) [31].

The 23 LTTRs mentioned above were present in different bacteria and *Citrobacter* is the genus that shares more LTTRs with *Salmonella*, supporting previous work showing that members of the *Citrobacter* genus resemble *Salmonella* more than other genera in the *Enterobacteriales* order [37] (Fig 3).

*S.* Typhi LTTR orthologous were also identified in different genera of the *Enterobacteriales* order such as *Escherichia, Enterobacter*, *Klebsiella*, *Shigella*, *Raoultella, Cronobacter*, *Kosokonia, Erwinia*, *Pectobacterium, Yersinia* and in *Hafnia* (Fig 3). The LTTRs mainly presented in these bacteria in descending order correspond to CysB, GcvA, OxyR, ArgP, YhaJ and IlvY [20–2326] (Fig 3).

In general, these data show that some LTTRs are specific and unique to a bacterial genus. Other LTTRs are disseminated in more genera, and there are LTTRs that are widely dispersed in nature.

## The LTTR CysB evolve as the core genome of *Enterobacteriales*

CysB consists of a conserved helix turn motif in the N terminal region of 56 aa, a linker helix of 26 aa and a variable C-terminal coinducer binding domain of 222 aa. CysB usually function as transcriptional activators of *cys* genes involved in sulfur metabolism and cysteine synthesis. Furthermore, CysB is also involved in iron metabolism, biofilm formation, swarming motility, the type III secretion system (T3SS) and is essential for the pathogenicity of *R. solanacearum* toward different host plants [38–44].

Since CysB is fundamental for essential biological process, form part of the *Salmonella* LTTR core (Fig 2) and are present in 68 genera (Fig 3 and 4), we analyzed whether this LTTR evolved as part of the core genome of *Enterobacteriales*. Molecular evolution analysis show that the evolutionary tree topology obtained with CysB, shows sufficient resolution to discriminate organisms at least at the genus level, suggesting that this gene has evolved in parallel within each group. We can also observe that branch lengths are not constant across the tree, which indicates different evolutionary rates that are likely associated with environmental adaptation events, or with the specific regulatory function of this protein within each genus. These results suggest that CysB can be considered as an additional phylogenetical tool, at least in the *Enterobacteriales* order (Fig 5).

A maximum likelihood phylogenetic tree based on CysB shows that this LTTR can group closely related genera into monophyletic clades. The colors show the phylogenetic groups. Bootstrap values support the phylogenetic relationships of CysB in *Enterobacteriales*. To generate the dendrogram CysB protein sequences from 68 genera were obtained and the species tree was obtained using the Best-fit model LG + R4 chosen according to Bayesian Information Criterion.

The result shows that CysB evolved like the genome core of the *Enterobacteriales* order. Furthermore, we analyzed the evolution of the LTTRs OxyR, ArgP, GcvA and IlvY, but these have a lower resolution for forming consistent clades. Thus, only CysB show the potential to separate these organisms at the taxonomic level of genus (Fig 5).

## Discussion

Transcriptional regulation is the main mechanism of bacterial adaptation to the environment. DNA-binding transcription regulators are pivotal components of the transcriptional regulatory network that typically sense diverse stimuli, recognize and bind to target genes to give an appropriate response to a given condition. Different families including the LTTR family of transcriptional regulators are widely distributed in bacteria and LTTRs controls the expression of genes involved in central metabolism, amino acid biosynthesis, cell division, fermentation, photosynthesis, nitrogen fixation, oxidative stress, quorum sensing, virulence, motility, and transport of Na + . Therefore LTTRs are essential in Gram-negative microorganisms [7,45].

In this work we describe that *S. enterica* (*S.* Typhi) have more LTTRs than *S. arizonae, diarizonae*, *houtenae, indica, salamae* and *S. bongori*. In this regard, is reported that the number of LTTRs depends on the diversity of environments occupied by the organism. The organism that presents the largest number of LTTRs is the β-proteobacteria *Burkholderia lata*, with 264 proteins [46]. This bacterium inhabits soil, water, flowers and is also found in patients with cystic fibrosis and nosocomial infections. In contrast, the endosymbiont of carpenter ants, the γ-proteobacteria *Blochmannia floridanus*, only contains one LTTR, and is limited to one habitat [47]. These examples show that the number of LTTRs is related to habitat variety. Another possible explanation about LTTR abundance is that the LTTR number depends on the genome size [7]. In this respect the genomes of *S. enterica* (*S.* Typhi)*, S. arizonae, diarizonae, houtenae, indica, salamae* and *S. bongori* are around 4.8 Mb, showing that in the *Salmonella* genus the LTTR number is independent of the genome size.

The data reported in this work also shows that in the case of exclusive human pathogens, such as *S.* Typhi the LTTR number is 41. Interestingly in other human pathogen such as *Mycobacterium tuberculosis* the LTTR number is decreased to only four [48] and in *Helicobacter pylori* LTTRs are absent [7]. These data show that even in intracellular human pathogens the LTTR number is different, furthermore these observations also show that in different pathogens, the recombination process such as gene duplication to generate new transcriptional regulators [49,50] can be higher or with lower frequency. Thus, in *Salmonella* LTTR generation by gene duplication and the maintenance of these duplicates may be is reflected in the different numbers of LTTRs in the *Salmonella* genus.

Another explanation is that the number of LTTRs in *Salmonella* varies because their regulatory cascades involve multiple LTTRs. In this regard we previously publish that multiple LTTRs are involved in specific biological process such as biofilm formation, bile resistance, and porin synthesis in *S.* Typhi [51]. Furthermore, in other organisms also multiple LTTRs are involved in swarming, tolerance to copper, zinc and to oxidative stress [52]. Determining why there are different numbers of LTTRs in the genus *Salmonella,* is a prosperous avenue of research that will undoubtedly generate novel and relevant information regarding the biology of these transcriptional factors in *Salmonella.*

In this report the *Salmonella* LTTR core was established. The LTTR core is defined as the set of LTTRs observed in all 4541 *Salmonella* genomes analyzed. Future analysis is needed to determine whether these LTTRs are fundamental in the free-living and pathogenic lifestyles of *Salmonella*. In this regard, individual mutants in each of the 90 LTTRs in the nitrogen fixing bacteria *Sinorhizobium meliloti* were able to survive in different media [53]. Perhaps the high number of these transcriptional factors allows bacteria to survive in specific conditions. In this sense in *Vibrio cholerae* the presence of two OxyR proteins was reported [54], supporting that the presence of multiple LTTRs could allow bacteria to replace specific LTTR absence.

The analysis of LTTR distribution shows that 23 *Salmonella* LTTRs are present in multiple microorganisms including *Citrobacter*, *Escherichia, Enterobacter*, *Klebsiella*, *Shigella*, *Raoultella, Cronobacter*, *Kosokonia, Erwinia*, *Pectobacterium, Yersinia* and in *Hafnia* (Figs 3, 4). Interestingly all these bacteria contain the LTTR CysB, and this protein presents an evolution similar at the genus which they belong. Therefore, there are LTTRs that evolve as fundamental genes as CysB and others that are required for specific functions. In this regard, the LTTR pool mainly present in *S. enterica* subspecies *enterica* provides different capabilities to these pathogens for survival in different compartments of the hosts. For example, LTTRs present in *Salmonella enterica* such as RipR are utilized to overcome itaconic acid stress in macrophages [55]. Another LTTR fundamental in *S.* Typhimurium virulence is YhjC. The deletion of YhjC reduced the replication ability of *S.* Typhimurium in macrophages and decreased the colonization of *S.* Typhimurium in mouse systemic organs (liver and spleen). The contribution of these LTTR to *Salmonella* virulence is indirect since they control multiple virulence genes [56]. Another LTTR that is indirectly related with virulence is LeuO*,* whose expression results in repression of the *Salmonella* pathogenic island I genes and inhibits *Salmonella* invasion of epithelial cells [57]. Another accessory LTTR is STM0435 that interacts with c-di-GMP. The c-di-GMP-STM0435 complex regulates flagellar synthesis, allowing *Salmonella* to survive in the host [58]. The list of the LTTR accessory pool for survival in the host or indirectly involved in virulence include SpvR. *Salmonella* serovars, such as *S.* Typhimurium, *S. enterica* subspecie *enterica* serovar Choleraesuis, Enteritidis and

Dublin harbors the *spv* genes on a large plasmid. The *spv* locus greatly enhances virulence in experimental infections of mice and is crucial for growth of the bacteria in the liver and spleen and SpvR is fundamental in the control of the *spv* locus [59]. Thus, there is a LTTR pool fundamental for specific functions such as virulence and pathogenesis.

Finally, we draw attention to the LTTRs exclusive to the *Salmonella* genus that are hypothetical proteins, as well as to the specific LTTRs of *S. enterica* (*S.* Typhi). These deserve special attention, since they are fundamental to understanding the physiological functions of these proteins in *Salmonella* with the aim of preventing, controlling or eradicating typhoid fever, diarrhea, gastroenteritis, bacteremia, endovascular infections and focal infections produced by *Salmonella* in children and adults, not only in developing countries but also in the industrialized world.

## Supporting information

**S1 Table. Table 1 supplemental material.** Accession numbers, organism names, and NCBI Taxonomy IDs of genomes. (XLSX)

## Acknowledgments

We would like to thank M. Dunn for stimulating discussions and critical reading. M. Fernández-Mora, F. J. Santana and A. Vazquez for scientific suggestions.

## Author contributions

**Conceptualization:** Dávila S, Gama-Martínez Y, Orozco R, Hernandez-Lucas I.

**Data curation:** Dávila S, Gama-Martínez Y, Orozco R, Rivera-Ramírez A.

**Formal analysis:** Dávila S, Gama-Martínez Y, Hernandez-Lucas I.

**Funding acquisition:** Hernandez-Lucas I.

**Investigation:** Dávila S, Gama-Martínez Y, Hernandez-Lucas I.

**Methodology:** Dávila S.

**Project administration:** Hernandez-Lucas I.

**Resources:** Hernandez-Lucas I.

**Software:** Rivera-Ramírez A.

**Supervision:** Dávila S, Orozco R, Hernandez VM, Hernandez-Lucas I.

**Validation:** Dávila S, Gama-Martínez Y, Rivera-Ramírez A, Hernandez VM, Hernandez-Lucas I.

**Visualization:** Dávila S, Rivera-Ramírez A, Hernandez VM.

**Writing – original draft:** Hernandez-Lucas I.

**Writing – review & editing:** Dávila S, Gama-Martínez Y, Hernandez VM, Hernandez-Lucas I.

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
