## [Decision Letter · Decision Letter 0]

12 Jul 2025

Dear Dr. Hernandez Lucas,

Thank you for submitting your manuscript to PLOS ONE. After careful consideration, we feel that it has merit but does not fully meet PLOS ONE’s publication criteria as it currently stands. Therefore, we invite you to submit a revised version of the manuscript that addresses the points raised during the review process.

**Kindly revise the manuscript by addressing the comments provided by me and the reviewers.**plosone@plos.org . A rebuttal letter that responds to each point raised by the academic editor and reviewer(s). You should upload this letter as a separate file labeled 'Response to Reviewers'.A marked-up copy of your manuscript that highlights changes made to the original version. You should upload this as a separate file labeled 'Revised Manuscript with Track Changes'.An unmarked version of your revised paper without tracked changes. You should upload this as a separate file labeled 'Manuscript'.

We look forward to receiving your revised manuscript.

Kind regards,

Priyanka Sharma

Academic Editor

PLOS ONE

**Journal Requirements:**

1. When submitting your revision, we need you to address these additional requirements. Please ensure that your manuscript meets PLOS ONE's style requirements, including those for file naming. The PLOS ONE style templates can be found at https://journals.plos.org/plosone/s/file?id=wjVg/PLOSOne_formatting_sample_main_body.pdf and https://journals.plos.org/plosone/s/file?id=ba62/PLOSOne_formatting_sample_title_authors_affiliations.pdf 2. Thank you for stating the following financial disclosure: UNAMDGAPA/UNAM (IN203621-IN202224)CONAHCYT (CF-2023-I-2079)    Please state what role the funders took in the study.  If the funders had no role, please state: "The funders had no role in study design, data collection and analysis, decision to publish, or preparation of the manuscript." If this statement is not correct you must amend it as needed. Please include this amended Role of Funder statement in your cover letter; we will change the online submission form on your behalf. 3. Thank you for stating the following in the Acknowledgments Section of your manuscript: We would like to thank M. Dunn for stimulating discussions and critical reading. M. Fernández-Mora, F. J. Santana and A. Vazquez for scientific suggestions. This work was supported by grants from: Dirección General de Asuntos del Personal Académico, DGAPA/UNAM (IN203621-IN202224) and CONAHCYT (CF-2023-I-2079) to I.H.L. We note that you have provided funding information that is not currently declared in your Funding Statement. However, funding information should not appear in the Acknowledgments section or other areas of your manuscript. We will only publish funding information present in the Funding Statement section of the online submission form. Please remove any funding-related text from the manuscript and let us know how you would like to update your Funding Statement. Currently, your Funding Statement reads as follows: UNAMDGAPA/UNAM (IN203621-IN202224)CONAHCYT (CF-2023-I-2079)   Please include your amended statements within your cover letter; we will change the online submission form on your behalf. 4. We note that your Data Availability Statement is currently as follows: All relevant data are within the manuscript and its Supporting Information files. Please confirm at this time whether or not your submission contains all raw data required to replicate the results of your study. Authors must share the “minimal data set” for their submission. PLOS defines the minimal data set to consist of the data required to replicate all study findings reported in the article, as well as related metadata and methods (https://journals.plos.org/plosone/s/data-availability#loc-minimal-data-set-definition). For example, authors should submit the following data: - The values behind the means, standard deviations and other measures reported;- The values used to build graphs;- The points extracted from images for analysis. Authors do not need to submit their entire data set if only a portion of the data was used in the reported study. If your submission does not contain these data, please either upload them as Supporting Information files or deposit them to a stable, public repository and provide us with the relevant URLs, DOIs, or accession numbers. For a list of recommended repositories, please see https://journals.plos.org/plosone/s/recommended-repositories. If there are ethical or legal restrictions on sharing a de-identified data set, please explain them in detail (e.g., data contain potentially sensitive information, data are owned by a third-party organization, etc.) and who has imposed them (e.g., an ethics committee). Please also provide contact information for a data access committee, ethics committee, or other institutional body to which data requests may be sent. If data are owned by a third party, please indicate how others may request data access. 5. We note that you have included the phrase “data not shown” in your manuscript. Unfortunately, this does not meet our data sharing requirements. PLOS does not permit references to inaccessible data. We require that authors provide all relevant data within the paper, Supporting Information files, or in an acceptable, public repository. Please add a citation to support this phrase or upload the data that corresponds with these findings to a stable repository (such as Figshare or Dryad) and provide and URLs, DOIs, or accession numbers that may be used to access these data. Or, if the data are not a core part of the research being presented in your study, we ask that you remove the phrase that refers to these data.

**Additional Editor Comments:**

Dear Authors,

You have addressed an important organism causing a major public health problem. I appreciate the workflow of study. I have few comments provided below. Please submit a revised version including the comments from editor and reviewers.

1. You have stated , "A phyloT phylogenetic tree of species based on ribosomal proteins was obtained from the NCBI genome taxonomy database (GTDb).". Please provide the proper link or reference for the readers.

2. You found no difference between the LTTRs in S. Typhi and other less virulent serotypes. How will you explain the significance of LTTRs now? Please provide more extensive discussion.

3. The quality of images needs to be improved. Please save the files with higher resolution.

Reviewers' comments:

Reviewer's Responses to Questions

**Comments to the Author**

1. Is the manuscript technically sound, and do the data support the conclusions?

Reviewer #1: Partly

Reviewer #2: Yes

Reviewer #3: Yes

2. Has the statistical analysis been performed appropriately and rigorously?

Reviewer #1: No

Reviewer #2: Yes

Reviewer #3: N/A

3. Have the authors made all data underlying the findings in their manuscript fully available?

Reviewer #1: Yes

Reviewer #2: Yes

Reviewer #3: Yes

4. Is the manuscript presented in an intelligible fashion and written in standard English?

Reviewer #1: Yes

Reviewer #2: Yes

Reviewer #3: No

**Reviewer #1:**  1) The Materials and Methods section briefly states that LTTRs were obtained from the Salmonella CT18 genome. However, the manuscript does not clarify the methodology used for their identification. The authors should elaborate on the approach: What specific tools, databases, or bioinformatic pipelines were used? Overall, the Methods section can be more descriptive.

2) Figure 1 needs improved visualization. The authors may consider redesigning this figure to enhance readability, perhaps by using a color-coded heatmap or a presence/absence matrix with clear gridlines. This would help the audience better interpret and appreciate the comparative distribution of LTTRs across strains or species.

3) I think a graphical presentation of the LTTR distribution will help the manuscript. The text refers multiple times to the common versus exclusive presence of LTTRs among strains. This information would be more impactful if presented graphically. The authors could consider including pie charts, bar plots, or Venn diagrams to summarize the shared and unique LTTRs across the dataset.

4) The claim regarding CysB as an evolutionary marker appears somewhat overstated. Unlike ribosomal proteins, which are universally conserved and widely used in phylogenetic studies, CysB is not present across all bacterial taxa and seems to be largely restricted to Enterobacteriaceae. At best, CysB may serve as a functional evolutionary marker within specific niches or lineages.

5) Moreover, regulatory genes such as CysB are often subject to horizontal gene transfer, which can confound phylogenetic interpretations. What are authors view on this. Maybe the authors should address these caveats and clarify whether their use of "evolutionary marker" refers to functional divergence rather than strict lineage tracing.

**Reviewer #2:** This is a very interesting study in LysR-Type transcriptional regulators (LTTRs). Further research is needed, for better understanding the role of these proteins, in order to prevent and control infections caused by Salmonella, in adults and children.

**Reviewer #3:**  Dear Authors,

Thank you for submitting your manuscript to PLoS ONE. Your study presents an insightful and methodologically strong investigation into the evolution and distribution of LTTRs in Salmonellae and other bacterial genera. The identification of core and accessory LTTR pools has meaningful implications for understanding bacterial adaptability and gene regulation. However, to improve the manuscript, kindly consider the following;

• Please revise for grammatical accuracy and clarity, particularly in the abstract, introduction, and discussion. Some phrases are meaningless because of either too short or too long to understand.

• Avoid repetition between results and discussion; use the latter to provide synthesis and implications.

• Add clarifications to your methods, such as genome accession details and criteria for LTTR classification.

• Where you discuss conserved LTTRs (e.g., CysB), explicitly indicate why it may serve as a genomic marker.

• The name of bacterial spp. should be written in standard way of being italic, however, the same is missing in the text, this needs to be carefully revised.

Once these revisions are made, the manuscript will be suitable for publication. Thank you again for your contribution.

Best regards,

Muhammad Athar Abbas, DVM, PhD

**Do you want your identity to be public for this peer review?** For information about this choice, including consent withdrawal, please see our Privacy Policy

Reviewer #1: No

Reviewer #2: No

Reviewer #3: **Yes:** Muhammad Athar Abbas, DVM, PhD

---

## [Author Response · Author response to Decision Letter 1]

15 Sep 2025

Comments of reviewer 1

Reviewer #1: 1) The Materials and Methods section briefly states that LTTRs were obtained from the Salmonella CT18 genome. However, the manuscript does not clarify the methodology used for their identification. The authors should elaborate on the approach: What specific tools, databases, or bioinformatic pipelines were used? Overall, the Methods section can be more descriptive.

Response: The new version of the manuscript includes subtitles, tools, databases, genome accessions as well as a new understandable explanation of the material and methods used in this study. Thus, we modify the complete section of materials and methods as you suggested.

Materials and methods.

LTTR identification in Salmonella and bacterial genomes.

The PROSITE database was used to obtain LTTRs from the S. Typhi CT18 genome (ID 220341) (https://prosite.expasy.org/PS50931). The conserved domain accessions PF0126 and PF03466 were applied to verify the S. Typhi LTTR authenticity. Furthermore ClustalW (https://www.genome.jp/tools-bin/clustalw) sequence alignment showed that all the LTTRs identified in S. Typhi CT18 genome contain the winged helix DNA binding domain, the linker helix domain and periplasmic binding domain characteristic of the LTTR family (9). The resultant 41 LTTRs of S. Typhi (Fig. 1) were used as input query sequences to find orthologous proteins in prokaryotic genomes contained in the UniProt database. LTTR sequences were selected only if they shared ≥90% amino acid identity with the LTTRs of S. Typhi. Using this filtering strategy S. Typhi LTTR orthologs were identified in 33 genomes of Salmonella enterica subspecies enterica, in one genome of Salmonella bongori and in one genome each of the subspecies arizonae, diarizonae, houtenae, indica and salamae. Thus, S. Typhi LTTRs orthologous sequences in 39 genomes belonging to the Salmonella genus were identified. Orthologous LTTRs sequences of S. Typhi CT18 from other bacterial phyla including Enterobacteriaceae and Phyllobacteriaceae. were also obtained using this filtering strategy. The LTTR data obtained was used to generate Figs.1- 4.

LTTR distribution in Salmonella and bacterial genomes.

A phylogenetic tree of species based on NCBI taxonomy of ribosomal proteins using the software PhyloT (https://phylot.biobyte.de/) was constructed to facilitate the visualization of LTTR distribution in Enterobacteriaceae and Phyllobacteriaceae. The resulting tree was manipulated, annotated and visually enhanced using of phylogenetic tool iTOL (https://itol.embl.de/). In the supplemental material we include the Taxon Id of the genomes utilized to generate Figure 3.

Phylogenetic relationships of bacteria based on the LTTR CysB and ribosomal proteins.

To determine the phylogeny of the LTTR CysB in the broader context of bacterial species, a phylogenetic tree of these organisms based on CysB proteins was generated. A second phylogenetic tree of bacterial species based on ribosomal protein sequences using data from the Genome Taxonomy Database (GTDB) was also constructed. Specifically, a phylogenetic reconciliation was carried out between the CysB tree and the ribosomal species-level tree. For all these analyses, ClustalW (https://www.genome.jp/tools-bin/clustalw) was employed to perform multiple sequence alignments, reconstruct, and compare the CysB and ribosomal protein phylogenies. The final visualization of the phylogenetic reconciliation analysis between CysB and ribosomal proteins (Fig.4) was created through a combination of phylogenetic tools, including iTOL for tree editing, Inkscape (https://inkscape.org/) for image modification and refinement, and R (https://www.r-project.org/) for additional data handling and figure generation.

2) Figure 1 needs improved visualization. The authors may consider redesigning this figure to enhance readability, perhaps by using a color-coded heatmap or a presence/absence matrix with clear gridlines. This would help the audience better interpret and appreciate the comparative distribution of LTTRs across strains or species.

Response: Figure 1 has been redesigning. The new figure includes colors and clear gridlines as you suggested

3) I think a graphical presentation of the LTTR distribution will help the manuscript. The text refers multiple times to the common versus exclusive presence of LTTRs among strains. This information would be more impactful if presented graphically. The authors could consider including pie charts, bar plots, or Venn diagrams to summarize the shared and unique LTTRs across the dataset.

Response: As you suggested, a graphical representation (Fig 2) of the LTTR distribution in the Salmonella genus was included in the new version of the manuscript

4) The claim regarding CysB as an evolutionary marker appears somewhat overstated. Unlike ribosomal proteins, which are universally conserved and widely used in phylogenetic studies, CysB is not present across all bacterial taxa and seems to be largely restricted to Enterobacteriaceae. At best, CysB may serve as a functional evolutionary marker within specific niches or lineages.

Response: In the new version of the manuscript, we modify the text about CysB as evolutionary marker, instead we added this paragraph as you suggested

We propose to the LTTR CysB as an additional tool to be used in molecular phylogenetic studies in Enterobacteriaceae.

5) Moreover, regulatory genes such as CysB are often subject to horizontal gene transfer, which can confound phylogenetic interpretations. What are authors view on this. Maybe the authors should address these caveats and clarify whether their use of "evolutionary marker" refers to functional divergence rather than strict lineage tracing.

Response: We are agreeing with your comment, thus we just stated the role of CysB as an additional tool to be used in molecular phylogenetic studies in Enterobacteriaceae.

Comments of reviewer 2

Reviewer #2: This is a very interesting study in LysR-Type transcriptional regulators (LTTRs). Further research is needed, for better understanding the role of these proteins, in order to prevent and control infections caused by Salmonella, in adults and children.

Response: You are right, in this regard our group currently are working in the LTTR characterization for better understanding the role of these proteins, in order to prevent and control infections caused by Salmonella in adults and children.

Comments of reviewer 3

Reviewer #3: Dear Authors,

Thank you for submitting your manuscript to PLoS ONE. Your study presents an insightful and methodologically strong investigation into the evolution and distribution of LTTRs in Salmonellae and other bacterial genera. The identification of core and accessory LTTR pools has meaningful implications for understanding bacterial adaptability and gene regulation. However, to improve the manuscript, kindly consider the following;

• Please revise for grammatical accuracy and clarity, particularly in the abstract, introduction, and discussion. Some phrases are meaningless because of either too short or too long to understand.

Response. A native speaker English has reviewed the new version of the manuscript.

• Avoid repetition between results and discussion; use the latter to provide synthesis and implications.

Response. Corrected as you suggested

• Add clarifications to your methods, such as genome accession details and criteria for LTTR classification.

Response. The new version of the manuscript includes subtitles, tools, databases, genome accessions and criteria for LTTR classification as well as and understandable explanation of the material and methods used in this study. Thus, we modify the complete section of materials and methods as you suggested.

• The name of bacterial spp. should be written in standard way of being italic, however, the same is missing in the text, this needs to be carefully revised.

Response. In the new version of the manuscript all the bacterial spp are correctly written as you suggested.

Once these revisions are made, the manuscript will be suitable for publication. Thank you again for your contribution.

Response: All your suggestions and corrections of other two reviewers and of the editor were addressed. Thus, the new version of the manuscript is now greatly improved, and we are grateful to you and the reviewers for their insightful comments.

In the supplemental Material we include a Table “Bacterial genomes utilized to generated Figure 3”

---

## [Decision Letter · Decision Letter 1]

17 Nov 2025

Dear Dr. Hernandez Lucas,

Thank you for submitting your manuscript to PLOS ONE. After careful consideration, we feel that it has merit but does not fully meet PLOS ONE’s publication criteria as it currently stands. Therefore, we invite you to submit a revised version of the manuscript that addresses the points raised during the review process.

Please submit your revised manuscript in Jan 01 2026 11:59PM. If you will need more time than this to complete your revisions, please reply to this message or contact the journal office at plosone@plos.org . A rebuttal letter that responds to each point raised by the academic editor and reviewer(s). You should upload this letter as a separate file labeled 'Response to Reviewers'.A marked-up copy of your manuscript that highlights changes made to the original version. You should upload this as a separate file labeled 'Revised Manuscript with Track Changes'.An unmarked version of your revised paper without tracked changes. You should upload this as a separate file labeled 'Manuscript'.

We look forward to receiving your revised manuscript.

Kind regards,

Priyanka Sharma

Academic Editor

PLOS ONE

Journal Requirements:

Additional Editor Comments:

The authors present a study on the establishment of a Salmonella LTTR core, which is an interesting and potentially valuable contribution. The revised version shows substantial improvement compared with the previous submission, and several analyses appear promising. Before the manuscript can be considered for publication, however, a number of clarifications, linguistic improvements, and additional references are required to strengthen the scientific arguments and ensure clarity. The manuscript would also benefit from careful language editing, ideally with input from a native English speaker, as several sentences contain grammatical errors or unclear phrasing.

Below are specific comments that should be addressed in the revised draft:

Major and Minor Comments

1. Introduction

Please provide a clear explanation of cysB and its biological role in the introduction. The introduction should also more thoroughly discuss the role and significance of LTTRs in Salmonella. The rationale for selecting cysB is not clearly articulated and should be stated explicitly.

2. Lines 29–30

This sentence appears incomplete. Please revise to clearly convey the intended meaning.

3. Lines 70–71

Please rewrite this sentence for clarity and correct grammar.

4. Lines 94–97

The filtering strategy used in the analysis needs to be described more clearly and in greater detail.

5. Lines 100–102

Please revise this statement. A phylogenetic tree is not typically used to assess the distribution of LTTRs.

6. Lines 104–105

Please check and correct the grammar.

7. Line 114

The phrase “CysB and ribosomal protein phylogenies” is unclear. The term phylogeny may not be the most appropriate here. Please revise for scientific accuracy.

8. Lines 115–118

The methods used for phylogenetic tree generation, including bootstrap values and all relevant parameters, should be described more thoroughly.

9. There are several existing studies reporting ribosomal phylogenetic trees. Please cite the relevant literature and discuss how your findings compare, particularly regarding clade structure and similarity.

10. Please discuss why LTTRs appear to be present in only specific serotypes and relate these observations to existing literature on LTTR expression, bacterial biology, and pathogenicity.

11. Since LTTR expression was identified in public databases, consider evaluating whether LTTR profiles correlate with antimicrobial susceptibility, if such metadata are available.

12. Lines 124–126

You mention that the primary reason for studying LTTRs is the high morbidity of Salmonella Typhi. Please clarify this rationale and specifically state why LTTRs were chosen for this investigation.

13. Lines 147–148

Please provide references supporting the statement that certain subspecies adapt to hosts or environments with a reduced number of LTTRs.

14. Lines 149–152

This statement seems overly general. Please provide scientific justification explaining why LTTR reduction might occur.

15. Lines 146–157

Please provide references and additional detail regarding the hypothetical proteins mentioned. Clarify whether the LTTR core differs among species or serotypes.

16. Lines 163–164

Are LTTR cores or cassettes present in other bacterial taxa? If so, please describe which genes are involved and how they compare to the Salmonella LTTR core, with appropriate references.

Additionally, discuss whether the specific LTTR distribution might contribute to serotype-specific pathogenic behaviors.

17. Lines 185–186

Please clarify the meaning of “Salmonella taxonomic group.”

18. Lines 190–191

The heading “LTTRs of the Salmonella genus are present in Enterobacteriaceae Phyllobacteriaceae” is confusing, since Salmonella itself belongs to Enterobacteriaceae. Consider revising to something like “Presence of LTTRs in Other Enterobacteriaceae Members” to improve clarity.

Lines 192–194: Please rewrite this section to convey the findings clearly.

19. Lines 202–203

Please clarify what is meant by “between 13 and 14.”

20. Line 206

Please revise the phrase “common evolutionary story of LTTR evolution” for clarity and scientific accuracy.

21. Lines 211–213

Please rewrite this statement for clarity and list the pathogens in a scientifically appropriate manner.

22. Lines 214–216

Please specify which bacteria share this particular order of LTTRs, and clarify whether this applies to all bacteria listed in lines 213–214.

23. Lines 218–221

The sentence contains repeated phrasing (“were obtained”). Please revise for clarity and grammatical correctness.

24. Lines 221–222

Please refer to the correct table number in the text.

25. Line 228

The term “Essential genome core” is not standard. Please use established terminology from phylogenomics.

26. Line 236

Please clarify how the presence of both variable and conserved regions is “fundamental for biological processes,” as stated.

27. Line 239

Please correct the grammar.

28. Lines 243–245

Are there references showing that any specific gene can serve as a phylogenetic classifier? In Salmonella, even MLST using seven housekeeping genes has limited discriminatory power. Please justify the claim that cysB can be used in this way, including supporting references or comparative examples.

29. Line 247

Please clarify the meaning of “Reconciliation” in this context.

30. Line 251

Please provide the species and accession numbers for sequences downloaded from each genus.

31. Please clarify the overall conclusion and practical relevance of your findings related to cysB.

32. The manuscript touches on transcriptional regulation by LTTRs but does not clearly explain its biological implications. Please expand on what LTTRs control and why this is relevant.

33. LTTRs are important for evolution and transcriptional regulation; however, this is not reflected in the discussion. Please elaborate and cite relevant literature.

34. Lines 275–278

The statement is unclear regarding whether LTTR number is dependent or independent of genome size. Please clarify and cite supporting references.

35. Line 298

When discussing genetic redundancy, please provide quantitative or descriptive information on how similar the LTTRs are.

36. Line 302

If LTTRs similar to Salmonella LTTRs are present in other species, please clarify this section to avoid implying that LTTRs are exclusive to Salmonella. Revise the wording to accurately describe how LTTRs are shared across species and how they differ.

37. Supplementary Table 1

Please clarify whether the listed numbers represent genome accession IDs.

38. Figures are currently low resolution and difficult to evaluate. Please provide high-quality, high-resolution figures in the revised submission.

Reviewers' comments:

Reviewer's Responses to Questions

**Comments to the Author**

Reviewer #1: All comments have been addressed

Reviewer #2: All comments have been addressed

2. Is the manuscript technically sound, and do the data support the conclusions?

Reviewer #1: Partly

Reviewer #2: Yes

3. Has the statistical analysis been performed appropriately and rigorously?

Reviewer #1: Yes

Reviewer #2: Yes

4. Have the authors made all data underlying the findings in their manuscript fully available?

Reviewer #1: Yes

Reviewer #2: Yes

5. Is the manuscript presented in an intelligible fashion and written in standard English?

Reviewer #1: Yes

Reviewer #2: Yes

Reviewer #1: (No Response)

Reviewer #2: This is a very interesting study in LysR-Type transcriptional regulators (LTTRs). Further research is needed, for better understanding the role of these proteins, in order to prevent and control infections caused by Salmonella, in adults and children. I have no further comments to add.

**Do you want your identity to be public for this peer review?** For information about this choice, including consent withdrawal, please see our Privacy Policy

Reviewer #1: No

Reviewer #2: No

---

## [Author Response · Author response to Decision Letter 2]

30 Jan 2026

+The authors present a study on the establishment of a Salmonella LTTR core, which is an interesting and potentially valuable contribution. The revised version shows substantial improvement compared with the previous submission, and several analyses appear promising. Before the manuscript can be considered for publication, however, a number of clarifications, linguistic improvements, and additional references are required to strengthen the scientific arguments and ensure clarity. The manuscript would also benefit from careful language editing, ideally with input from a native English speaker, as several sentences contain grammatical errors or unclear phrasing.

Below are specific comments that should be addressed in the revised draft:

Major and Minor Comments

Introduction

Please provide a clear explanation of cysB and its biological role in the introduction. The introduction should also more thoroughly discuss the role and significance of LTTRs in Salmonella. The rationale for selecting cysB is not clearly articulated and should be stated explicitly. Corrected as you suggested.

Lines 29–30

This sentence appears incomplete. Please revise to clearly convey the intended meaning. Corrected as you suggested

Lines 70–71

Please rewrite this sentence for clarity and correct grammar. Corrected as you suggested

Lines 94–97

The filtering strategy used in the analysis needs to be described more clearly and in greater detail. Corrected as you suggested

Lines 100–102

Please revise this statement. A phylogenetic tree is not typically used to assess the distribution of LTTRs. This phrase was eliminated as you suggested

Lines 104–105

Please check and correct the grammar. Corrected as you suggested

Line 114

The phrase “CysB and ribosomal protein phylogenies” is unclear. The term phylogeny may not be the most appropriate here. Please revise for scientific accuracy. This phrase was eliminated as you suggested

Lines 115–118

The methods used for phylogenetic tree generation, including bootstrap values and all relevant parameters, should be described more thoroughly. Corrected as you suggested

There are several existing studies reporting ribosomal phylogenetic trees. Please cite the relevant literature and discuss how your findings compare, particularly regarding clade structure and similarity. The comparison of ribosomal and CysB phylogenetic trees was eliminated, since ribosomal proteins are widely distributed phylogenetic marker in most of the bacteria reported so far, in the case of CysB it is only present in few bacteria taxa and in this work, we show that CysB is only able to separate clades at the genus level in the Enterobacterial order. Thus, CysB can be considered as an additional phylogenetical tool only in Enterobacteriales.

Please discuss why LTTRs appear to be present in only specific serotypes and relate these observations to existing literature on LTTR expression, bacterial biology, and pathogenicity. The LTTR numbers are variable in distinct pathogens, maybe the duplication events and the maintenance of these duplicates in specific Salmonella serotypes are the reason because the LTTR number is different in this genus. Another explanation is that the number of LTTRs in Salmonella varies because their regulatory cascades involve multiple LTTRs, in this regard we previously publish that multiple LTTRs are involved in specific biological process such as biofilm formation, bile resistance and porin synthesis (https://doi.org/10.1371/journal.pone.0338130), even more, in other organisms also multiple LTTRs are involved in swarming, tolerance to copper, zinc and to oxidative stress https://doi.org/10.1016/j.ijbiomac.2024.132315. This discussion is included in the new version of the manuscript as you suggested

Lines 124–126

You mention that the primary reason for studying LTTRs is the high morbidity of Salmonella Typhi. Please clarify this rationale and specifically state why LTTRs were chosen for this investigation. Corrected as you suggested

Lines 147–148

Please provide references supporting the statement that certain subspecies adapt to hosts or environments with a reduced number of LTTRs. This statement seems overly general. Please provide scientific justification explaining why LTTR reduction might occur. We are agree with you, the statement is very general, that's why it was eliminated, and the new version of the manuscript only describe the results obtained.

Lines 146–157

Please provide references and additional detail regarding the hypothetical proteins mentioned. References about the hypothetical proteins mentioned were included in the new version of the manuscript as you suggested. Clarify whether the LTTR core differs among species or serotypes. The LTTR core is the same in all the serotypes described, this information was clarified in the new version of the manuscript as you suggested

Lines 163–164

Are LTTR cores or cassettes present in other bacterial taxa? If so, please describe which genes are involved and how they compare to the Salmonella LTTR core, with appropriate references. Your question is the subject of another research project. In this manuscript, we only focus on the Salmonella genus.

Additionally, discuss whether the specific LTTR distribution might contribute to serotype-specific pathogenic behaviors. The LTTR distribution does not correspond to pathogenic behavior, since in S. Typhi LTTR number is 41 and in other human pathogen such as Mycobacterium tuberculosis the LTTR number decreased to only four and in Helicobacter pylori LTTRs are absent. These data show that even in intracellular human pathogens the LTTR number is different. Determining why there are different numbers of LTTRs in the genus Salmonella, is a prosperous avenue of research that will undoubtedly generate novel and relevant information regarding the biology of these transcriptional factors in Salmonella. This information as well as other data were included in the new version of the manuscript, as you suggested.

Lines 185–186

Please clarify the meaning of “Salmonella taxonomic group.” This phrase was removed to avoid confusion.

Lines 190–191

The heading “LTTRs of the Salmonella genus are present in Enterobacteriaceae Phyllobacteriaceae” is confusing, since Salmonella itself belongs to Enterobacteriaceae. Consider revising to something like “Presence of LTTRs in Other Enterobacteriaceae Members” to improve clarity. The heading was modified as you suggested.

Lines 192–194: Please rewrite this section to convey the findings clearly. Corrected as you suggested

Lines 202–203

Please clarify what is meant by “between 13 and 14.” We eliminate between 13 and 14 to avoid confusion, as you suggested

Line 206

Please revise the phrase “common evolutionary story of LTTR evolution” for clarity and scientific accuracy. We remove this phrase to avoid confusion, as you suggested

Lines 211–213

Please rewrite this statement for clarity and list the pathogens in a scientifically appropriate manner. We remove part of this phrase to avoid confusion

Lines 214–216

Please specify which bacteria share this particular order of LTTRs, and clarify whether this applies to all bacteria listed in lines 213–214. We modify this paragraph for clarity, as you suggested.

Lines 218–221

The sentence contains repeated phrasing (“were obtained”). Please revise for clarity and grammatical correctness. Were obtained was eliminated, and the new version of the manuscript only contain once time were obtained

Lines 221–222

Please refer to the correct table number in the text. Corrected as you suggested

Line 228

The term “Essential genome core” is not standard. Please use established terminology from phylogenomics. The subtitle was rewrite in the new version of the manuscript

Line 236

Please clarify how the presence of both variable and conserved regions is “fundamental for biological processes,” as stated. This part of the paragraph was eliminated to avoid confusion.

Line 239

Please correct the grammar. Corrected as you suggested.

Lines 243–245

Are there references showing that any specific gene can serve as a phylogenetic classifier? In Salmonella, even MLST using seven housekeeping genes has limited discriminatory power. Please justify the claim that cysB can be used in this way, including supporting references or comparative examples. You are right. Therefore, we only mention that CysB orthologous sequences separate clades at the genus level, suggesting that CysB has evolved in parallel with the corresponding lineage. Thus, CysB can be considered as an additional phylogenetical tool in the Enterobacteriales order

Line 247

Please clarify the meaning of “Reconciliation” in this context. We remove the comparison of ribosomal proteins and CysB. Therefore, the word reconciliation was eliminated

Line 251

Please provide the species and accession numbers for sequences downloaded from each genus. The accession numbers of the sequences utilized in this work were provided in the supplementary table 1

Please clarify the overall conclusion and practical relevance of your findings related to cysB. Corrected as you suggested.

The manuscript touches on transcriptional regulation by LTTRs but does not clearly explain its biological implications. Please expand on what LTTRs control and why this is relevant. Corrected as you suggested.

LTTRs are important for evolution and transcriptional regulation; however, this is not reflected in the discussion. Please elaborate and cite relevant literature. Corrected as you suggested.

Lines 275–278

The statement is unclear regarding whether LTTR number is dependent or independent of genome size. Please clarify and cite supporting references. Corrected as you suggested

Line 298

When discussing genetic redundancy, please provide quantitative or descriptive information on how similar the LTTRs are. We eliminated the phrase genetic redundancy to avoid confusion

Line 302

If LTTRs similar to Salmonella LTTRs are present in other species, please clarify this section to avoid implying that LTTRs are exclusive to Salmonella. Revise the wording to accurately describe how LTTRs are shared across species and how they differ. Corrected as you suggested

Supplementary Table 1

Please clarify whether the listed numbers represent genome accession IDs. Corrected as you suggested

Figures are currently low resolution and difficult to evaluate. Please provide high-quality, high-resolution figures in the revised submission. Corrected as you suggested

---

## [Editor Report · Decision Letter 2]

18 Feb 2026

Distribution and evolution of the LysR-Type transcriptional regulators of the Salmonella genus.

PONE-D-25-23217R2

Dear Authors,

We’re pleased to inform you that your manuscript has been judged scientifically suitable for publication and will be formally accepted for publication once it meets all outstanding technical requirements.

Kind regards,

Priyanka Sharma

Academic Editor

PLOS One

Additional Editor Comments (optional):

Thank you for addressing all the comments. I am happy to accept this manuscript for publication.

---

## [Editor Report · Acceptance letter]

PONE-D-25-23217R2

PLOS One

Dear Dr. I,

I'm pleased to inform you that your manuscript has been deemed suitable for publication in PLOS One. Congratulations! Your manuscript is now being handed over to our production team.

Kind regards,

on behalf of

Dr. Priyanka Sharma

Academic Editor

PLOS One